# Learngene Tells You How to Customize: Task-Aware Parameter Prediction at Flexible Scales

## Abstract

Reducing serving costs and latency is a fundamental challenge for deploying large-scale models in business applications. To cope with this demand, the *Learngene* framework encapsulates shareable information from large models into a compact unit called a learngene. This unit serves to initialize downstream models, enabling them to inherit the knowledge from the large model efficiently, hopefully diminishing deployment expenses. However, existing learngene methods are constrained by their strong dependence on the architecture of large model and overlook the features of target tasks, resulting in suboptimal adaptability of downstream models to deployment requirements. In this paper, we present *Task-Aware Learngene* (**TAL**), a novel method based on graph hypernetworks that predicts model parameters conditioned on desired model scales and task-specific characteristics. Extensive experiments demonstrate that TAL effectively scales model initialization parameters, selectively utilizes shareable information pertinent to target tasks, and consistently outperforms random initialization and existing parameter prediction methods. Furthermore, TAL exhibits promising transfer learning capabilities for unseen tasks, underscoring its effectiveness in condensing large model knowledge while being aware of downstream requirements.

## 1 Introduction

Large-scale foundation models Dosovitskiy (2020); Radford et al. (2021) have revolutionized multi-task learning by enabling the fine-tuning of a single model across various downstream tasks Dosovitskiy (2020); Liu et al. (2021). This paradigm rests on the belief that the shareable information among different tasks allows foundation models to provide effective parameter initialization for arbitrary tasks. However, fine-tuning typically necessitates maintaining the entire model's parameters and updating them for a specialized task Mahabadi et al. (2021), which usually comes at a significant cost. Furthermore, the size of the fine-tuned model may not always align with the scale of the downstream task Houlsby et al. (2019), posing unnecessary computational burdens, particularly in real-time deployments with stringent latency or memory footprint constraints. These challenges have sparked great interest in developing techniques that target specific issues left by the pre-training and fine-tuning paradigm, such as parameter-efficient fine-tuning methods Houlsby et al. (2019); Li & Liang (2021); Lester et al. (2021) and model compression methods Liu et al. (2017); Jiao et al. (2019); Touvron et al. (2021).

Beyond conventional strategies, a new concept of *learngene* Wang et al. (2022) offers an alternative parameter initialization scheme. Inspired by the intergenerational transfer of information in biology, a learngene philosophically acts as a functional unit that encapsulates shareable information from a pretrained model (referred to as the *ancestry model* in learngene-based methods). Instead of replicating the entire pretrained model to obtain a parameter initialization, downstream models (termed *descendant models*) *inherit* the learngene, that is, initialized by learngene in some way. Thus the Learngene framework is expected to confer many advantages, such as efficient knowledge transfer and adaptability to descendant models of flexible scales, however, the exploration for its central challenges in the representation and extraction of learngene remains in its early stages.

The initial learngene work Wang et al. (2022) posited that the memorization of shareable information can be localized, that is, certain neurons exhibit greater generalizability. Consequently, the approach aims to identify these neurons and directly extract them, along with their trained parameters, to form a learngene.

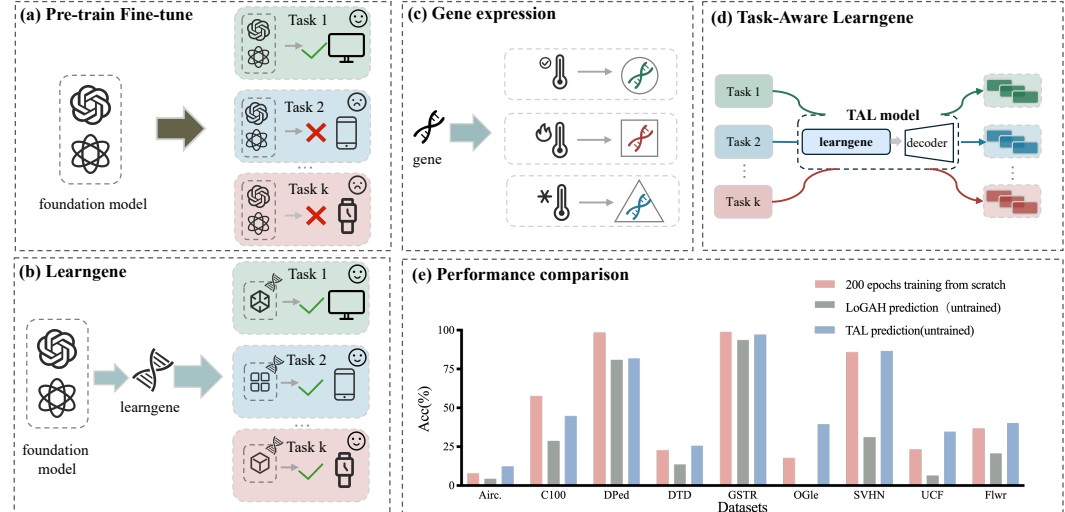

Figure 1: (a) Pre-train fine-tune approach does not necessarily work for many real-world application scenarios. (b) Learngene method can better adapt to different task scenarios. (c) In biology, genes exhibit different expressions under different environments, *e.g.*, temperatures. (d) Task-Aware Learngene (TAL) generates task-specific model parameters for different tasks. (e) Untrained descendant models initialized with TAL outperform those trained for 200 epochs with random initialization across different datasets.

By combining the learngene with an arbitrary number of randomly initialized neurons, the descendant model is effectively initialized. More advanced efforts Shi et al. (2024); Xia et al. (2024b;a), focused on the Transformer architecture, seeking to identify a single module that can approximate a linear mapping to all modules in the ancestry network, thereby serving as a learngene. This kind of approach enables a more compact representation of information across the entire network while ensuring that the process of flexibly initializing the descendant model does not dilute the efficacy of the learngene.

However, current methods heuristically translate shareable information into shareable parameters, thus learngene only providing parameter initialization information to descendant models with specific architectures — typically those identical to the ancestry model — which leads to downstream models that remain insufficiently flexible. Recently, Graph HyperNetworks (GHNs) Zhang et al. (2018); Knyazev et al. (2021) have demonstrated the ability of encoding the network architectures. A GHN is composed of an encoder that learn the compact latent representations from different network architectures and a decoder that then uses those latent representations to reconstruct the networks. Once trained, the hypernetwork's weights encode the network architectures, which to some extent resonates with the need for learngene to integrate structural information, but its applicability in the Learngene framework is not straightforward. More importantly, it would be preferable if the process of using the learngene to initialize a descendant model *not only supports variable architectures with scaling but also customizes the parameters based on the specific downstream task it targets*.

In this paper, we explore a GHN-based learngene method to encode shareable information to predict initial parameters for descendant models across flexible scales. Our approach enables the learngene to dynamically adjust its parameter initialization in response to the requirements of descendant models, thereby significantly enhancing the scalability of existing learngene methods. By leveraging a hypernetwork, we achieve an end-to-end mechanism for the representation and transfer of shareable knowledge, seamlessly integrating structural information into the parameter initialization process. Taking one step beyond existing learngene methods, we introduce a fine-tuning procedure that imbues the learngene with task-specific awareness, allowing it to modulate the expression of shared knowledge tailored to targeted downstream tasks. Crucially, our approach is the first to explore dual customization of descendant models based on both scale and task characteristics, yielding a significantly improved quality-cost trade-off for information sharing. We systematically investigate the effectiveness of TAL. Extensive experiments show the superiority of TAL. For example, descendant models initialized with TAL outperform those initialized using LoGAH Zhou et al. (2024) by an average of 24.39% across Decathlon datasets Rebuffi et al. (2017).

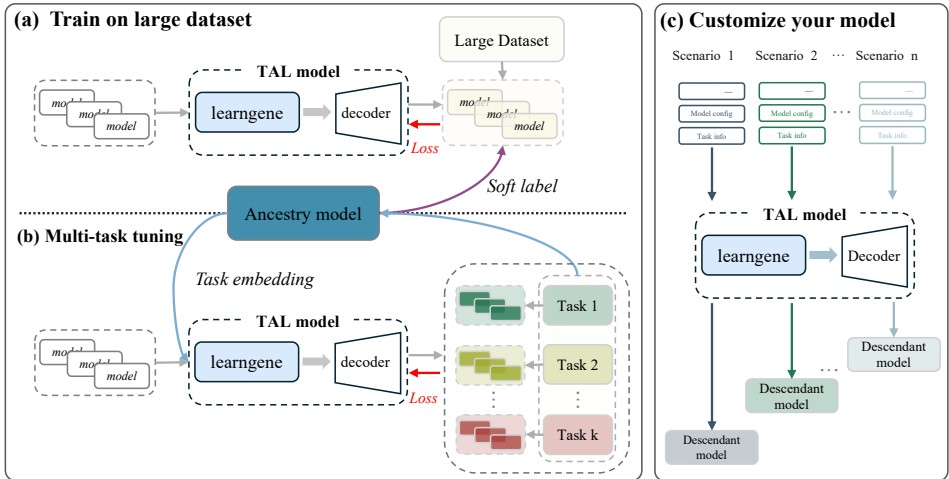

Figure 2: (a) Training TAL model on a large dataset under the guidance of a large-scale foundation model (ancestry model). (b) Tuning TAL model to multiple tasks. (c) Customizing task-specific Descendant models with flexible scale based on new task scenarios.

Our main contributions are summarized follows: (1) We propose a new Learngene method that uses an encoder-decoder structure to generate descendant models, enabling greater flexibility in scales. (2) We design a task-aware learngene that transfers task-specific knowledge across different tasks and predicts model parameters based on desired model scales and task-specific characteristics. (3) Extensive experiments demonstrate that the effectiveness of TAL, *e.g.*, compared to training from scratch, descendant models initialized with TAL can achieve better performance while reducing huge training cost.

## 2 PRELIMINARIES

**Graph HyperNetworks.** Graph HyperNetworks method (GHNs) is originally proposed for neural architecture search. The input of GHN $H_D(\theta)$ is a computational graph $f^G$ of a neural network $f$ and the output of GHN is the parameters of the model $\mathbf{w}_{\text{pred}} = H_D(f^G;\theta)$.

In Knyazev et al. (2021), GHN $H_D$ is trained by SGD over $M$ training architectures $\{f_a^G\}_{a=1}^M$ and $N$ training data samples $\{x_j, y_j\}_{j=1}^N$ on the following optimization problem:

$$\underset{\theta}{\text{argmin}} \frac{1}{NM} \sum_{j=1}^{N} \sum_{a=1}^{M} \mathcal{L}(f_a(x_j; H_D(f_a^G;\theta)), y_j), \tag{1}$$

when training GHN $H_D(\theta)$, a meta-batch of $m$ training architectures is sampled as input for GHN. Meanwhile, a mini-batch of $n$ training datas $\mathbf{x}$ is sampled and fed into the parameter-predicted $m$ architectures to get $m \times n$ predictions $y_hat$. The cross-entropy loss $\mathcal{L}$ is computed between $y_hat$ and ground truth labels $y$ of $\mathbf{x}$ for classification tasks. Afterward, the loss is back-propagated to update the parameters $\theta$ of $H_D$ by gradient descent.

The latest GHN work improves the decoder part of the model, introducing LoGAH Zhou et al. (2024), a low-rank Graph HyperNetwork (GHN), which avoids the redundancy of multiple copies of small parameter blocks when predicting large-scale parameters, greatly improving the prediction ability and scalability of model parameters.

In GHN-1/2 Knyazev et al. (2021) and GHN-3 Knyazev et al. (2023), training architectures are sampled from DeepNets-1M a dataset of 1 million architectures Knyazev et al. (2021), in LoGAH works, to generate Transformer models for classification tasks, the build ViTs-1K datasets Zhou et al. (2024), which contains 1K different ViT-style computational graphs. In this work, We use the ViTs-1K model dataset to train a HyperNetwork to generate ViT models of various sizes and shapes.

## 3 METHOD

Fig. 2 illustrates the overall pipeline of Task-Aware Learngene (TAL). First, we train the TAL model on a large dataset under the guidance of an ancestry model to transfer knowledge. Then, we tune the trained TAL model on multi-task datasets to effectively filter and convey the task-specific knowledge from the ancestry model to descendant models. Finally, the trained TAL model can predicting descendant model parameters for various tasks, even unseen ones and supports model customization at different scales. Next, we introduce the problem definition for TAL.

### 3.1 PROBLEM DEFINITION

We are committed to addressing the challenges that large-scale foundation models face when adapting to downstream tasks. First, fine-tuning typically necessitates maintaining the entire model's parameters and updating them for a specialized task, which usually comes at a significant cost. Additionally, in resource-constrained environments, the large memory footprint of the foundation model can hinder its deployment and using large models for small-scale tasks may result in unnecessary computational burdens and resource wastage.

TAL seeks to inherit knowledge from ancestry models, while selectively expressing in specific tasks. This enables descendant models initialized with TAL to adapt to various tasks in resource-limited environments. These descendant models not only inherit task-specific and shared knowledge from the large model but also have the flexibility to adjust their size according to task requirements, thereby better accommodating diverse task scenarios.

### 3.2 TASK-AWARE LEARNGENE

**TAL model structure and components.** In the TAL, we adopt encoder-decoder structure for model parameters prediction Knyazev et al. (2023); Zhou et al. (2024). We refer to the encoder part of the TAL model as learngene because it first inherits knowledge from the ancestry model and then transfers task-specific knowledge based on different tasks. Specifically, learngene receives both model configuration through model computational graph and task information. Based on task information, learngene can filter out task-specific knowledge which previously inherited from the ancestry model and inject it into model computational graphs, thereby producing task-specific computational graphs.

The architecture of learngene is shown in Fig. 3. Inspired by Perez et al. (2018); Oreshkin et al. (2018), we introduce a task hypernet $h$ that processes task information to dynamically generate parameters for the task-specific layer (TSL), which is implemented as a simple MLP. Then task-specific layer acts on the model computational graph, transferring task information to the it.

In this process, task information is passed in the form of a task embedding $\{I_\tau\}_{\tau=1}^T$ for each task, which is generated by the ancestry model through the average feature extraction of the task images Vu et al. (2020).

The task hypernet $h$ generates task bias parameters $\gamma_\tau$, and $\beta_\tau$ of the task-specific layer.

$$(\gamma_\tau, \beta_\tau) := h(I_\tau) = (W^\gamma, W^\beta) I_\tau, \tag{2}$$

where $W^\gamma \in \mathbb{R}^{h \times t}$ and $W^\beta \in \mathbb{R}^{h \times t}$.

The task-specific layers apply these bias parameters to the model's computation graph using the following formula:

$$f_\tau^G = \gamma_\tau \times f^G + \beta_\tau \tag{3}$$

The task-specific model computational graph generated by learngene is then passed to the decoder Zhou et al. (2024), which decodes the graph to generate the model parameters.

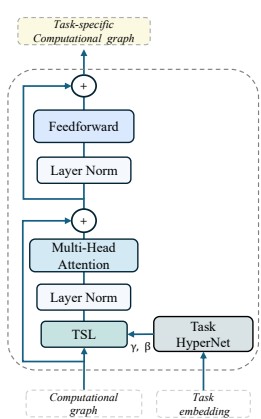

Figure 3: The learngene is based on a transformer architecture and consists of a stack of transformer blocks.

| Model | Training state | RandInit | LoGAH v1 | **TAL** |
|---|---|---|---|---|
| 12-layer ViT-Tiny | Untrained | 0.10 | 22.79 | **26.20** |
| | Trained | **64.98** | 62.53 | 63.03 |
| 12-layer ViT-Small | Untrained | 0.10 | 16.78 | **23.28** |
| | Trained | 64.65 | 65.41 | **66.61** |

Table 1: Performance of descendant models on ImageNet-1K initialized with RandInit, LoGAH v1 and TAL, after 75 epochs of training for all initialization methods.

**Train TAL model on a large dataset.**  In order to inherit the knowledge of the ancestry model, we first train the TAL model on a large dataset under the guidance of the ancestry model. Since this process does not involve multi-task training, we do not use the TSL module in learngene. We adopt the method from Shu et al. (2021), converting features extracted by the ancestry model from the images into a probability distribution map. For the training models of the TAL, we also apply the same method to obtain their feature probability distributions and compute the KL divergence between those of the ancestry model. This function is denoted as:

$$\mathcal{L}_{\text{aux}} = \text{KL}(\text{softmax}(E_{\text{train}}M)||\text{softmax}(E_{\text{anc}})), \tag{4}$$

where $E_{\text{train}}$ and $E_{\text{anc}}$ refer to the encoders' output of the training model and ancestry model respectively. The matrix $M \in \mathbb{R}^{d \times d'}$, which transforms the output dimension $d$ of $E_{\text{train}}$ to match the output dimension $d'$ of $E_{\text{anc}}$, like the parameters of other parts of the training model, the transformation matrix's parameters are predicted directly by TAL model.

Considering loss between training model's predicted label and ground truth label:

$$\mathcal{L}_{cls} = \text{CE}(y_c, f_{\text{train}}(x)), \tag{5}$$

where $f_{\text{train}}(x)$ represents the training model's predicted label of input image data $x$ and $y_c$ denotes the ground truth label belonging to category c. Then, while one model is used as training data for TAL model, the total training loss is computed as follows:

$$\mathcal{L} = \alpha\mathcal{L}_{aux} + (1-\alpha)\mathcal{L}_{cls}, \tag{6}$$

Through this process, the TAL model can inherit and utilize the vast amount of domain knowledge already learned in the ancestry model, enabling models initialized by TAL model to handle complex tasks.

**Tuning TAL model under multi-task setting.**  We then tune the TAL model on multiple tasks, leveraging task information to enable learngene to generate task-specific computation graphs, thereby decoding the model parameters tailored to each task. We formulate the loss function for this part of the TAL model's training. Given the data from a set of tasks $\{\mathcal{D}_\tau\}_{\tau=1}^T$, here $T$ is the total number of tasks and $\mathcal{D}_\tau = \{(x_i^\tau, y_i^\tau)\}_{i=1}^{N_\tau}$ shows the training data for $\tau$-th task with $N_\tau$ samples.

Assuming there is a TAL model $H_D(\theta)$ parameterized by $\theta$ that computes the output the parameters of the model $\mathbf{w}_{\text{pred}} = H_D(f^G; \theta)$ for input computational graph $f^G$ of a neural network. In multi-task setting, TAL model is trained by SGD over $M$ training models $\{f_a^G\}_{a=1}^M$ and T training tasks $\{\mathcal{D}_\tau\}_{\tau=1}^T$ on the following optimization problem:

$$\arg\min_\theta \frac{1}{TM} \sum_{\tau=1}^{T} \sum_{a=1}^{M} \sum_{(x_\tau^j, y_\tau^j) \in \mathcal{D}_\tau} w_\tau \mathcal{L}(f_a(x_\tau^j; H_D(f_a^G; \theta)), y_\tau^j), \tag{7}$$

where $\mathcal{L}$ is typically the cross-entropy loss and $w_\tau$ shows the sampling weight for $\tau$-th task.

Multi-task training allows the descendant models predicted by TAL to inherit task-specific knowledge filtered by learngene from the ancestry model, as well as the shared knowledge across tasks.

**Customize descendant models.**  After training on a large dataset and tuning on multiple tasks, the TAL model can provide task-specific, variable-sized models for both seen and unseen tasks. By simply passing the required model configuration and task information to the TAL model, one can instantly obtain well-initialized model parameters tailored to the task at hand.

## 4  EXPERIMENTS

In this section, we employ our proposed Task-Adaptive Learning (TAL) method to predict model parameters for diverse tasks at varying scales. First, we evaluate TAL's capability to predict model parameters using

| Model | Method | Airc. | C100 | DPed | DTD | GSTR | OGle | SVHN | UCF | Flwr | Avg |
|---|---|---|---|---|---|---|---|---|---|---|---|
| 3-Ti | GHN-3 | 3.12 | 34.06 | 85.41 | 6.38 | 87,77 | 0.06 | 10.00 | 2.25 | 7.06 | 26.23 |
| | LoGAH v1 | 2.58 | 29.16 | 78.83 | 8.30 | 92.12 | 0.26 | 17.32 | 3.94 | 6.96 | 26.61 |
| | LoGAH v2 | 2.07 | **40.91** | **83.06** | 11.22 | 91.15 | 0.20 | 26.04 | 24.44 | 9.22 | 32.03 |
| | **TAL** | **6.69** | 39.53 | 79.83 | **22.55** | **94.86** | **28.37** | **83.31** | **28.33** | **33.82** | **46.37** |
| 6-Ti | GHN-3 | 3.24 | 35.19 | 87.72 | 6.86 | 89.12 | 0.06 | 10.00 | 3.43 | 10.88 | 27.39 |
| | LoGAH v1 | 3.21 | 46.33 | 81.19 | 9.2 | 96.82 | 0.31 | 20.82 | 5.02 | 8.33 | 30.14 |
| | LoGAH v2 | 2.31 | 48.18 | **86.79** | 23.83 | 94.02 | 0.15 | 26.54 | 24.85 | 12.35 | 35.45 |
| | **TAL** | **17.43** | **48.95** | 85.87 | **28.3** | **99.25** | **48.98** | **89.85** | **38.63** | **45.69** | **55.88** |
| 12-Ti | GHN-3 | 3.15 | 31.12 | 85.63 | 6.86 | 85.02 | 0.06 | 10.00 | 3.07 | 9.90 | 26.09 |
| | LoGAH v1 | 3.15 | 33.80 | 79.86 | 9.41 | 96.52 | 0.18 | 20.56 | 5.53 | 9.12 | 28.68 |
| | LoGAH v2 | 1.32 | **46.64** | **86.38** | 4.04 | 93.83 | 0.18 | 25.10 | 23.92 | 11.27 | 32.52 |
| | **TAL** | **16.71** | 44.72 | 84.76 | **28.03** | **99.15** | **28.11** | **88.87** | **39.81** | **45.00** | **52.80** |
| 3-S | GHN-3 | 2.91 | 35.75 | 86.77 | 6.70 | 87.68 | 0.06 | 10.00 | 2.61 | 10.39 | 26.99 |
| | LoGAH v1 | 3.15 | 44.95 | 80.66 | 9.36 | 96.35 | 0.28 | 19.91 | 5.33 | 8.73 | 29.86 |
| | LoGAH v2 | 2.16 | 46.93 | **86.43** | 21.86 | 94.11 | 0.11 | 26.33 | 25.2 | 12.55 | 35.08 |
| | **TAL** | **17.16** | **47.25** | 83.78 | **27.23** | **98.93** | **47.75** | **87.83** | **38.32** | **44.31** | **54.73** |
| 6-S | GHN-3 | 3.12 | 35.30 | 86.80 | 7.34 | 90.05 | 0.06 | 10.00 | 2.36 | 12.75 | 27.53 |
| | LoGAH v1 | 3.18 | 45.93 | 80.60 | 9.95 | 96.97 | 0.26 | 21.04 | 5.38 | 8.63 | 30.22 |
| | LoGAH v2 | 2.43 | 48.43 | **85.95** | 24.2 | 94.95 | 0.20 | 25.16 | 25.51 | 15.59 | 35.82 |
| | **TAL** | **17.85** | **49.88** | 83.91 | **28.67** | **99.30** | **50.29** | **89.82** | **40.83** | **46.47** | **56.34** |
| 12-S | GHN-3 | 2.64 | 5.55 | 84.30 | 7.23 | 84.39 | 0.06 | 10.00 | 1.74 | 9.51 | 22.82 |
| | LoGAH v1 | 2.64 | 35.77 | 80.39 | 8.24 | 96.81 | 0.18 | 20.66 | 4.30 | 6.57 | 28.40 |
| | LoGAH v2 | 1.98 | 12.90 | **85.36** | 15.74 | 94.66 | 0.22 | 16.14 | 21.77 | 11.86 | 28.96 |
| | **TAL** | **17.16** | **45.63** | 82.16 | **27.87** | **99.18** | **39.66** | **88.55** | **40.98** | **45.10** | **54.03** |

Table 2: Performance of untrained descendant models on Decathlon tasks initialized with LoGAH v1, LoGAH v2 and TAL.

| Model | Method | Airc. | C100 | DPed | DTD | GSTR | OGle | SVHN | UCF | Flwr | Avg |
|---|---|---|---|---|---|---|---|---|---|---|---|
| 3-Ti | RandInit | 7.41 | 53.33 | 97.93 | 22.93 | 98.58 | 18.31 | 87.10 | 22.08 | 37.94 | 49.51 |
| | GHN-3 | 5.04 | 47.39 | 97.18 | 21.44 | 98.15 | 1.54 | 10.00 | 22.44 | 30.29 | 37.05 |
| | LoGAH v1 | 8.70 | **55.42** | 96.16 | 20.59 | 98.78 | 17.85 | 83.85 | 23.36 | 32.65 | 48.60 |
| | LoGAH v2 | 9.66 | 53.05 | 98.03 | 28.09 | 98.74 | 20.59 | 85.49 | 43.65 | 32.25 | 52.21 |
| | **TAL** | **19.8** | 54.89 | **98.61** | **30.21** | **99.57** | **63.57** | **90.38** | **48.1** | **47.84** | **61.44** |
| 6-S | RandInit | 8.79 | 61.80 | 98.67 | 22.61 | 99.11 | 17.13 | 84.78 | 24.39 | 35.39 | 50.30 |
| | GHN-3 | 5.16 | 53.43 | 97.94 | 19.15 | 97.99 | 9.46 | 10.00 | 23.72 | 30.98 | 38.65 |
| | LoGAH v1 | 9.12 | **62.25** | 97.41 | 18.99 | 98.69 | 25.62 | 86.35 | 32.27 | 33.43 | 51.56 |
| | LoGAH v2 | 10.38 | 59.80 | **98.74** | **31.54** | 99.09 | 24.91 | 86.08 | **52.20** | 42.45 | 56.13 |
| | **TAL** | **19.17** | 61.03 | 98.69 | 30.90 | **99.85** | **63.68** | **91.77** | 48.41 | **49.61** | **62.57** |

Table 3: Performance of trained descendant models on Decathlon tasks initialized with RandInit, LoGAH v1, LoGAH v2 and TAL. For descendant models initialized with RandInit, accuracy is reported after 200 epochs of training for each task, while for models initialized with other methods, trained for 100 epochs.

both training tasks and previously unseen tasks. Subsequently, we conduct an ablation study to investigate various contributing factors and present visualization experiments to demonstrate the effectiveness of TAL.

## 4.1 EXPERIMENTAL SETUP

**Datasets.** We use the ViTs-1K dataset Zhou et al. (2024), which contains 1000 different ViT-style computational graphs. We conduct experiments on Visual Domain Decathlon Challenge Rebuffi et al. (2017), containing 10 datasets: (1) ImageNet-1K (**IN-1K**) Russakovsky et al. (2015), (2) CIFAR-100 (**C100**) Krizhevsky et al. (2009), (3) Aircraft(**Airc.**) Maji et al. (2013), (4) Daimler pedestrian classification (**DPed**) Munder & Gavrila (2006), (5) Describable textures (**DTD**) Cimpoi et al. (2014), (6) German traffic signs (**GSTR**) Stallkamp et al. (2012), (7) Omniglot (**OGlt**) Lake et al. (2015), (8) **SVHN** Netzer et al. (2011), (9) UCF101 Dynamic Images (**UCF**) Soomro et al. (2012), (10) Flowers102 (**Flwr**) Nilsback & Zisserman (2008). The multi-task training with Decathlon datasets in the experiment means using nine other tasks besides ImageNet-1K. For a detailed description to the dataset, see the appendixA.1.

**Baselines.** We compare TAL with GHN-3 Knyazev et al. (2023) and the latest LoGAH method Zhou et al. (2024), which improves the design of the decoder and significantly enhances the initialized models' performance. Besides, we use the most efficient LoGAH-small model. Following the training strategy of LoGAH, we train individual LoGAH models for each task, referring to as LoGAH v1. Additionally we

| Dataset | Model | Training epochs | RandInit | LoGAH v1 | TAL |
|---|---|---|---|---|---|
| **fashion MNIST** | 3-Ti | 5 | 82.71 | 87.56 | **88.86** |
| | | 100 | 89.41 | **91.08** | 90.99 |
| | 6-S | 5 | 83.66 | 87.78 | **89.78** |
| | | 100 | 88.98 | 90.68 | **91.56** |
| **Fer2013** | 3-Ti | 5 | 29.20 | 42.57 | **47.06** |
| | | 100 | 59.91 | 60.60 | **61.41** |
| | 6-S | 5 | 30.00 | 32.57 | **49.99** |
| | | 100 | 62.55 | 61.94 | **65.09** |
| **HAM10000** | 3-Ti | 5 | 83.09 | **87.56** | 86.59 |
| | | 100 | 97.46 | **97.71** | **97.71** |
| | 6-S | 5 | 82.13 | 89.61 | **91.18** |
| | | 100 | 97.70 | 97.83 | **97.95** |

Table 4: Performance of descendant models on unseen tasks initialized with RandInit, LoGAH v1 (trained on IN-1K) and TAL, after 5 and 100 epochs of training for each task.

first train a GHN model on ImageNet-1K using the LoGAH v1 and subsequently tune it for each specific task, which is referred to LoGAH v2.

**Sampling tasks.** During multi-task training, we sample tasks using conventional temperature-based sampling Raffel et al. (2020) with a temperature of $T = 2$ for all methods. Tasks are sampled proportionally to $p_\tau^{1/T}$, where $p_\tau = \frac{N_\tau}{\sum_{i=1}^T N_i}$ and $N_\tau$ represents the number of training samples for the $\tau$-th task.

**Training Details.** For experiments on ImageNet-1K, the hypernets are trained for 75 epochs using the LoGAH v1 and the TAL. For datasets in the Decathlon Challenge, the GHN models are trained for 300 and 100 epochs using the LoGAH v1 and the LoGAH v2, respectively. For Decathlon's multi-task training, the TAL model is trained for 100 epochs using the TAL. All models are trained using automatic mixed precision in PyTorch, with a cosine annealing learning rate schedule starting at lr $= 3e-4$, weight decay $\lambda = 1e-2$ and predicted parameter regularization $\gamma = 3e-5$ Knyazev et al. (2023). For TAL, we use ViT-Base Dosovitskiy (2020) as the ancestry model.

## 4.2 MAIN RESULTS

**TAL achieves better performance than LoGAH trained on ImageNet-1K.** We evaluate the performance of the TAL on the ImageNet-1K. As shown in Tab. 1, the untrained descendant models, structured as 12-layer ViT-Tiny and ViT-Small, initialized using the TAL, outperform those initialized with LoGAH v1 by **3.41%** and **6.50%** on ImageNet-1K, respectively. Furthermore, after 75 epochs training, the two descendant models initialized with TAL achieve higher accuracy, with an increase of **0.50%** and **1.20%** compared to those initialized with LoGAH v1. These results show that TAL can effectively inherit and utilize the knowledge already learned in the ancestry model.

**Descendant models initialized with TAL demonstrate strong performance without any training on Decathlon tasks.** We compare TAL with random initialization (RandInit) and the LoGAH v1 and v2 methods, utilizing descendant models of various sizes, including ViT-Tiny and ViT-Small with 3, 6 and 12 layers, to evaluate the effectiveness of different initialization methods on Decathlon tasks. The selected descendant models are not part of the ViTs-1K model dataset and are therefore unseen by any of the methods. As shown in Tab. 2, untrained descendant models initialized with TAL outperform the LoGAH v1 by **24.39%** and the LoGAH v2 by **20.06%** across Decathlon tasks.

**Descendant models initialized with TAL converge faster and outperform those initialized by the LoGAH method during the training process.** We select two descendant models for further evaluation: a 3-layer ViT-tiny (3-Ti) and a 6-layer ViT-small (6-S). Tab. 3 shows that the 3-Ti model initialized with the TAL, achieve average accuracy improvements of **11.93%**, **12.84%** and **9.23%** across Decathlon tasks compared to the RandInit, LoGAH v1 and LoGAH v2, respectively. Similarly, the 6-S model, initialized with TAL and subsequently trained, demonstrate average accuracy improvements of **12.27%**, **11.01%** and **6.14%** across the same tasks when compared to the RandInit, LoGAH v1 and LoGAH v2. Using TAL, we can effectively inherit the knowledge of the ancestry model and extract shared knowledge across multiple tasks, thereby providing high-quality initialization parameters for different tasks. As shown in Tab. 2 and 3, descendant models initialized with TAL, even without training, outperform models that were initialized

| Method | Airc. | C100 | DPed | DTD | GSTR | OGle | SVHN | UCF | Flwr | Avg |
|---|---|---|---|---|---|---|---|---|---|---|
| *Single-Task Training* | | | | | | | | | | |
| LoGAH v1 | 2.99 | 39.32 | 80.26 | 9.08 | 95.93 | 0.25 | 20.05 | 4.92 | 8.06 | 28.98 |
| LoGAH v2 | 2.05 | 40.67 | **85.66** | 16.82 | 93.79 | 0.18 | 24.22 | 24.28 | 12.14 | 33.31 |
| *Multi-Task Training* | | | | | | | | | | |
| LoGAH v3 | 1.64 | 19.91 | 81.39 | 16.36 | 92.16 | 10.69 | 79.67 | 3.94 | 4.52 | 34.48 |
| LoGAH v4 | 12.69 | 43.17 | 80.60 | 23.36 | **98.95** | **51.07** | 87.52 | 4.86 | 40.88 | 49.23 |
| **TAL** | **15.50** | **46.00** | 83.39 | **27.11** | 98.45 | 40.53 | **88.04** | 37.82 | 43.40 | **53.37** |

Table 5: Performance of untrained descendant models on Decathlon tasks initialized with LoGAH v1–v4 and TAL.

| Method | Train on IN-1K | STT | MTT |
|---|---|---|---|
| LoGAH v1 | ✗ | ✓ | ✗ |
| LoGAH v2 | ✓ | ✓ | ✗ |
| LoGAH v3 | ✗ | ✗ | ✓ |
| LoGAH v4 | ✓ | ✗ | ✓ |

Table 6: Comparison of different LoGAH methods, indicating whether they use IN-1K for training and whether they employ Single-Task Training (STT) or Multi-Task Training (MTT).

| Method | anc-net | TSL | Avg Acc |
|---|---|---|---|
| TAL(w/o TSL) | ✓ | ✗ | 40.71 |
| TAL(w/o ans-net) | ✗ | ✓ | 46.80 |
| **TAL** | ✓ | ✓ | 53.37 |

Table 7: Performance of untrained descendant models on Decathlon datasets initialized with TAL without using TSL in learngene or ancestry model(ans-net) on Decathlon datasets.

with other methods with further training. For example, for the 6-layer ViT-small, the untrained descendant models initialized using TAL, achieve an average accuracy that are **6.04%**, **4.78%** and **0.21%** higher than trained models that are initialized using RandInit, LoGAH v1 and LoGAH v2 on the Decathlon datasets.

**TAL presents superior parameter prediction ability across unseen tasks.** We evaluate the TAL on a broader set of unseen datasets. Specifically, we use three datasets from distinct fields: Fashion MNIST Xiao et al. (2017), a dataset of fashion item images; FER2013 Goodfellow et al. (2013), a facial expression recognition dataset; and HAM10000m Tschandl et al. (2018), a medical dataset for the classification of skin lesions. We train the LoGAH v1 on ImageNet-1K to obtain a GHN model that predicts model parameters for transfer to unseen tasks. In contrast, our TAL directly initializes descendant models based on tasks, allowing for more task-specific adaptation. Tab. 4 shows that descendant models initialized with TAL converge faster and achieve higher test accuracy on unseen downstream tasks. For example, compared to the LoGAH v1, the 6-S models initialized with TAL demonstrate accuracy improvements of **0.88%**, **3.15%** and **0.12%** higher in accuracy for fashion MNIST, Fer2013 and HAM10000, respectively. Compared with RandInit, it is **2.58%**, **2.54%** and **0.25%** higher, respectively.

### 4.3 ANALYSIS AND ABLATION

In the main experiments, high-quality model initialization is shown to significantly accelerate convergence and improve final test accuracy. Therefore, in our analysis, we evaluate the performance of untrained descendant models on each dataset. For further analysis, we design two multi-task training methods based on LoGAH. First, we perform multi-task training of the GHN model directly on the nine tasks of the Decathlon challenge, referred to as LoGAH v3. Second, we first train the GHN model on ImageNet-1K and then tune it on Decathlon datasets for multi-task training, referred to as LoGAH v4. The differences between all LoGAH methods are detailed in the Tab. 6.

**TAL outperforms LoGAH v3.** As shown in Tab. 5, using LoGAH v3, the average accuracy of the descendant models on Decathlon tasks is **1.17%** higher than that of LoGAH v2. It demonstrates that multi-task training enables knowledge sharing between tasks, which can significantly improve the accuracy of descendant models. However, compared to TAL, the LoGAH v3 results in a **18.89%** lower average accuracy of descendant models across the Decathlon tasks.

**TAL outperforms LoGAH v4.** Tab. 5 shows TAL outperforms LoGAH v4 in almost all tasks. Specifically, descendant models initialized using TAL achieve higher test accuracy than those initialized with LoGAH v4 by **2.81%**, **2.83%**, **2.79%**, **3.75%**, **0.52%**, **32.96%**, **2.52%** on Air., C100, Died, DTD, SVHN, UCF and Flwr, respectively.

Task
information

model → *Computational graph* → **learngene** → *Task-specific Computational graph* → decoder → model

Figure 4: Learngene processes the model computational graph and task information to generate a task-specific computational graph, which is then decoded to generate the model parameters.

**The effect of ancestry model and learngene.** As shown in Tab. 7, for TAL(w/o TSL) we remove the TSL module in learngene and for TAL(w/o ans-net) we remove the guidance of the ancestry model. Descendant models initialized using TAL achieve higher test accuracy across Decathlon tasks by **12.66%** and **6.57%** compared to models initialized with TAL (w/o TSL) and TAL (w/o ans-net), respectively.

**Visualization of task-specific model computational graphs.** Fig. 4 illustrates the process of predicting model parameters using the TAL model. First, based on the model configuration, the model is compiled into a standardized computational graph and passed as input to the learngene. The learngene simultaneously receives task information and, by integrating the model configuration with the task information, generates a task-specific computational graph. This learned computational graph is then passed to the decoder, which decodes it to generate the model parameters for the specific task.

To verify the effectiveness of learngene in dynamically encoding the model computational graph under different task conditions, we visualize the output of learngene. We use 3-12 layers of ViT-small, a total of 10 descendant models and apply learngene to output their task-specific computational graphs on Decathlon Challenge datasets. We use the PCA Abdi & Williams (2010) method to map the high-dimensional features of the model computational graph to 2D space and visualize them. The result is shown in Fig. 5. As the task information changes, the model's learned computational graph exhibits a significant clustering effect, the learned computational graphs of the model for different tasks clearly cluster together in two-dimensional space. This indicates that learngene can effectively integrate task information while incorporating task information into the model computational graph.

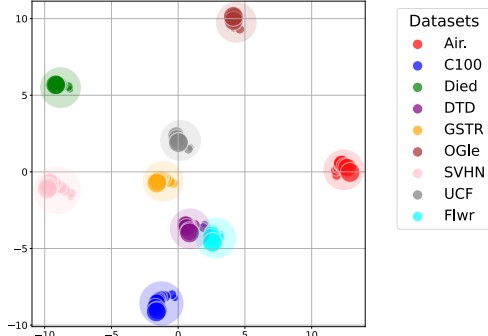

Figure 5: The computational graphs of all descendant models generated by learngene on the Decathlon datasets. Each point represents a model computational graph. Different colors denote different tasks and the size of the point corresponds to the model's scale, with larger points indicating larger models.

## 5 RELATED WORK

### 5.1 PARAMETER PREDICTION

Model parameter prediction is usually achieved through hypernetworks. A lot of research is devoted to extending the parameter prediction capabilities of hypernetworks to make them applicable to unseen model architectures or datasets. Among many methods, Graph HyperNetworks (GHN) Zhang et al. (2018); Knyazev et al. (2021) have garnered considerable attention due to their outstanding performance and high flexibility. GHN-2 Knyazev et al. (2021) and GHN-3 Knyazev et al. (2023) further improved the parameter prediction capabilities of GHN by improving the learning process of the model computation graph. The latest LoGAH method Zhou et al. (2024) introduces low-rank approximation (LoRA) technology, allowing GHN to predict the parameters of larger models using smaller hypernetworks. This progress has greatly improved the efficiency and ability of GHN in handling large-scale model parameter prediction tasks. Our task-aware learngene (TAL) incorporates modules to process task information, enabling a single TAL model to customize models of varying scales for different tasks.

## 5.2 LEARNGENE

The Learngene method Wang et al. (2022), inspired by biological gene inheritance, focuses on extracting compact components, known as Learngene, from large-scale pretrained models (referred to as ancestry models) and using them to initialize descendant models. Existing methods such as Vanilla-LG Wang et al. (2022), TLEG Xia et al. (2024b), Learngene Pool Shi et al. (2024) and SWS Xia et al. (2024a) employ different strategies to select and expand Learngene to build descendant models. In the Vanilla-LG method, the key layers of the ancestry model are screened and extracted as Learngene and are spliced together with randomly initialized layers to construct descendant models. The TLEG method employs a linear expansion of the layers of Learngene to initialize different descendant models. The Learngene Pool method refines a large-scale pretrained model into multiple small models, regarding the layers of these small models as Learngene instances, which are then spliced together to construct different descendant models. In task-aware learngene (TAL), learngene is no longer a sub block of the ancestry model, the encoder part of TAL model is regarded as learngene. Model generation becomes a process of encoding and decoding, with the ancestry model and multi-task common knowledge injected through the learngene. TAL model can handle both model and task information, initializing models of flexible scales for different tasks.

## 5.3 MULTI-TASK LEARNING

The goal of multitask learning is to train a model to perform well on multiple tasks simultaneously. It needs to solve multiple challenges Arivazhagan et al. (2019), such as catastrophic forgetting and deal with the disproportionate task sizes, resulting in overfitting in low-resource tasks and underfitting in high-resource tasks. An efficient multi-task learning method was proposed Mahabadi et al. (2021), using hypernetworks to achieve a common adapter between tasks. In existing work on model parameter prediction, the focus is often limited to predicting model parameters of varying sizes for a single task, without attempting to initialize models with different parameters for multiple tasks. Our work represents a significant advancement in the field, as it enables the prediction of parameters for different tasks and models of varying sizes.

## 6 CONCLUSION

In this paper, we propose a novel method based on graph hypernetworks called Task-Aware Learngene that predicts model parameters conditioned on desired model scales and task-specific characteristics. Experimental results on various datasets demonstrated the effectiveness of TAL's ability to predict parameters. Untrained descendant models initialized using TAL achieved significant improvements across various datasets compared to the previous LoGAH initialization methods. Remarkably, the accuracy of these untrained descendant models even surpassed the performance of models trained using other initialization methods.

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

# A   APPENDIX

## A.1   DECATHLON DATASET

The Visual Domain Decathlon Challenge tests the ability of visual recognition algorithms to handle images from different visual domains. It includes 10 datasets in total:

1. **ImageNet-1K** (IN-1K) is the largest dataset in the Decathlon Challenge, containing 1,000 categories and 1.2 million images.

2. **CIFAR-100** (C100) contains 60,000 $32 \times 32$ color images for 100 object categories.

3. **Aircraft** (Airc.) contains 100 images for each of 100 different aircraft model variants, such as the Boeing 737-400 and the Airbus A310.

4. **Daimler Pedestrian Classification** (DPed) consists of 50,000 grayscale pedestrian and non-pedestrian images, cropped and resized to $18 \times 36$ pixels.

5. **Describable Textures** (DTD) is a texture database consisting of 5,640 images, organized into 47 categories such as bubbly, cracked andmarbled.

6. **German Traffic Signs** (GTSRB) contains cropped images for 43 common traffic sign categories in different image resolutions.

7. **Omniglot** (OGlt) consists of 1,623 different handwritten characters from 50 unique alphabets.

8. **SVHN** is a real-world digit recognition dataset with around 70,000 $32 \times 32$ images.

9. **UCF101 Dynamic Images** (UCF) is an action recognition dataset of realistic human action videos, collected from YouTube. It contains 13,320 videos across 101 action categories. In the Decathlon Challenge, the videos are converted into images using Dynamic Image encoding, which summarizes each video into an image based on a ranking principle.

10. **Flowers102** (Flwr) is a fine-grained classification task with 102 flower categories from the UK, each consisting of 40 to 258 images.

The detailed statistics of the datasets can be found at `http://www.robots.ox.ac.uk/~vgg/decathlon/`.

## A.2 EXPERIMENT DETAILS

Here we present the detailed results of the analysis and ablation studies for LoGAH v3, LoGAH v4, TAL (w/o TSL), and TAL (w/o ans-ant) methods, as well as all descendant models on the Decathlon test.

| Model | Method | Airc. | C100 | DPed | DTD | GSTR | OGle | SVHN | UCF | Flwr | Avg |
|---|---|---|---|---|---|---|---|---|---|---|---|
| 3-Ti | LoGAH v3 | 1.35 | 17.87 | 80.27 | 15.8 | 88.24 | 10.29 | 78.63 | 3.12 | 2.65 | 33.14 |
| | LoGAH v4 | 12.72 | 42.42 | 78.93 | 23.14 | 98.81 | 44.19 | 86.07 | 5.02 | 39.71 | 47.89 |
| 6-Ti | LoGAH v3 | 1.77 | 19.84 | 79.35 | 16.44 | 91.35 | 10.75 | 79.07 | 4.15 | 4.80 | 34.17 |
| | LoGAH v4 | 12.81 | 44.27 | 84.49 | 24.04 | 98.95 | 55.05 | 87.79 | 5.12 | 40.59 | 50.35 |
| 12-Ti | LoGAH v3 | 1.44 | 19.58 | 79.78 | 16.12 | 91.53 | 10.63 | 79.50 | 3.69 | 4.71 | 34.11 |
| | LoGAH v4 | 12.36 | 40.06 | 65.39 | 21.33 | 98.92 | 44.1 | 87.80 | 4.20 | 36.95 | 45.68 |
| 3-S | LoGAH v3 | 1.56 | 20.12 | 83.06 | 16.76 | 93.08 | 10.26 | 80.06 | 4.30 | 4.22 | 34.82 |
| | LoGAH v4 | 12.81 | 44.38 | 84.10 | 24.47 | 98.97 | 52.2 | 87.28 | 5.12 | 43.14 | 50.27 |
| 6-S | LoGAH v3 | 1.83 | 21.66 | 83.32 | 16.60 | 94.87 | 12.01 | 80.34 | 4.25 | 5.00 | 35.54 |
| | LoGAH v4 | 13.14 | 45.16 | 87.04 | 24.04 | 99.11 | 57.01 | 88.33 | 5.17 | 43.43 | 51.38 |
| 12-S | LoGAH v3 | 1.86 | 20.41 | 82.53 | 16.44 | 93.87 | 10.21 | 80.43 | 4.10 | 5.78 | 35.07 |
| | LoGAH v4 | 12.30 | 42.74 | 83.64 | 23.14 | 98.98 | 53.87 | 87.87 | 4.51 | 41.46 | 49.83 |

Table 8: Performance of untrained descendant models on Decathlon tasks initialized with LoGAH v3, LoGAH v4.

| Model | Method | Airc. | C100 | DPed | DTD | GSTR | OGle | SVHN | UCF | Flwr | Avg |
|---|---|---|---|---|---|---|---|---|---|---|---|
| 3-Ti | TAL(w/o TSL) | 4.05 | 17.61 | 72.65 | 11.12 | 65.9 | 1.76 | 35.77 | 2.25 | 9.61 | 24.52 |
|  | TAL(w/o ans-net) | 3.39 | 49.92 | 92.74 | 20.80 | 99.39 | 0.18 | 90.2 | 11.37 | 50.69 | 46.52 |
| 6-Ti | TAL(w/o TSL) | 12.18 | 40.62 | 72.89 | 23.30 | 99.20 | 52.57 | 88.77 | 2.87 | 34.22 | 47.40 |
|  | TAL(w/o ans-net) | 3.27 | 49.86 | 92.38 | 20.05 | 99.35 | 0.17 | 90.34 | 12.09 | 51.18 | 46.52 |
| 12-Ti | TAL(w/o TSL) | 10.23 | 36.09 | 65.83 | 17.13 | 96.00 | 31.28 | 83.92 | 3.02 | 25.10 | 40.96 |
|  | TAL(w/o ans-net) | 2.88 | 49.45 | 93.01 | 21.22 | 99.31 | 0.15 | 90.43 | 12.19 | 51.08 | 46.64 |
| 3-S | TAL(w/o TSL) | 12.54 | 40.65 | 80.24 | 20.69 | 98.99 | 49.85 | 85.99 | 4.35 | 36.86 | 47.79 |
|  | TAL(w/o ans-net) | 2.88 | 50.76 | 92.76 | 21.97 | 99.35 | 0.09 | 90.51 | 11.53 | 52.35 | 46.91 |
| 6-S | TAL(w/o TSL) | 13.68 | 42.43 | 77.19 | 22.61 | 99.30 | 52.65 | 89.28 | 3.28 | 37.75 | 48.69 |
|  | TAL(w/o ans-net) | 3.12 | 50.91 | 93.10 | 22.29 | 99.29 | 0.14 | 90.57 | 12.81 | 52.25 | 47.16 |
| 12-S | TAL(w/o TSL) | 11.73 | 37.67 | 60.17 | 16.97 | 55.30 | 23.35 | 76.45 | 2.97 | 29.22 | 34.87 |
|  | TAL(w/o ans-net) | 2.88 | 50.86 | 93.52 | 21.76 | 99.34 | 0.12 | 90.64 | 12.19 | 52.16 | 47.05 |

Table 9: Performance of untrained descendant models on Decathlon tasks initialized with TAL(w/o TSL) and TAL(w/o ans-net).

