# OpenReview forum: "Learngene Tells You How to Customize: Task-Aware Parameter Prediction at Flexible Scales"
_ICLR.cc/2025/Conference — ICLR 2025 Conference Withdrawn Submission_

### Official Review · Reviewer_xAXT · 2024-10-27

**Soundness:** 2
**Presentation:** 3
**Contribution:** 3
**Rating:** 5
**Confidence:** 3

**Summary:**

This paper introduces an approach to address the cost and complexity of deploying large-scale models for various downstream tasks by creating a framework called Task-Aware Learngene (TAL). TAL builds on the learngene concept, which encapsulates transferable information from a large model into a compact, reusable unit. Unlike previous methods, TAL utilizes graph hypernetworks to predict model parameters that are both task-aware and scalable, thus enabling models to inherit large model knowledge with better adaptability and lower deployment costs. Through experiments, TAL shows good performance in initializing models compared to traditional methods like LoGAH, demonstrating improved accuracy across various datasets, quicker convergence, and adaptability to unseen tasks.

**Strengths:**

TAL provides a cost-effective means to transfer knowledge by creating a compact learngene unit, making it easier to deploy models with limited resources.
The task-specific customization through TAL’s graph hypernetwork enables the model to adapt to different downstream tasks, outperforming existing methods by a margin on unseen tasks.
TAL supports flexibility in model scaling, allowing the adjustment of model parameters based on specific task requirements. This scalability is particularly valuable for application scenarios.
TAL's initialization outperforms other methods like LoGAH across datasets, as evidenced by experiments showing TAL’s capability to provide higher-quality initialization parameters.

**Weaknesses:**

Overall, I personally like the idea of model customization with dynamic architectures and task-specific parameters. I have some concerns mainly regarding the experiments part.

The related works need improvement. Clearly stating the differences and contributions make the paper review evaluation better.

The baselines are limited. Comparing the latest one is good but diverse baselines from different aspects make the results stronger.

The datasets can be somewhat out-of-dated. Considering using challenging datasets would make the results and conclusions stronger, especially in an era of foundation models.

**Questions:**

Please refer to the weaknesses.

---

> ### Author Response · Authors · 2024-11-23
> **Response to reviewer  xAXT**
>
> We thank you for your reviews and address your concerns as follows.
> ### Q 1:
> The related works need improvement. Clearly stating the differences and contributions make the paper review evaluation better.
> ### A 1:
> We supplement this in the revised version on page 9 and 10. The added content as follows:
>
> Parameter Prediction:
> Our task-aware learngene (TAL) incorporates modules to process task information, enabling a single TAL model to customize models of varying scales for different tasks.
>
>
> Learngene:
> In task-aware learngene (TAL), learngene is no longer a sub block of the ancestry model, the encoder part of TAL model is regarded as learngene. Model generation becomes a process of encoding and decoding, with the ancestry model and multi-task common knowledge injected through the learngene. TAL model can handle both model and task information, initializing models of flexible scales for different tasks.
>
> ### Q 2:
> The baselines are limited. Comparing the latest one is good but diverse baselines from different aspects make the results stronger.
> ### A 2:
> We have added relevant experiments for the new baseline GHN-3 in the revised version on page 6.
> ### Q 3:
> The datasets can be somewhat out-of-dated. Considering using challenging datasets would make the results and conclusions stronger, especially in an era of foundation models.
> ### A 3:
> We train TAL-GPT2 on NLP tasks using the same approach employed for vision tasks, enabling parameter prediction for GPT-2 across various scales tailored to different tasks. Currently, the TAL-GPT2 model is still under training.

---

> > ### Comment · Reviewer_xAXT · 2024-11-26
> >
> > Thank you for your rebuttal. I find the idea interesting but believe the paper needs more work to meet the ICLR standard. I will keep my score for now and consider other reviewers' comments.

---

> > > ### Author Response · Authors · 2024-12-03
> > >
> > > Thanks for your response.
> > >
> > > We conduct extensive experiments by training TAL-GPT2 on multiple datasets: MRPC, COLA, RTE and IMDB. We compare two parameter prediction approaches - LoGAH and TAL - across different scales of GPT2-small models. The results demonstrated that TAL not only reduced training time by more than 50\% compared to LoGAH but also achieved superior performance across test tasks.
> > >
> > > For performance evaluation, we employ task-specific metrics: both accuracy and F1 recall for MRPC, Matthews correlation coefficient for COLA, and accuracy for RTE and IMDB. The experimental results are detailed as follows.
> > >
> > > | Model | Method | MRPC | COLA | RTE | IMDB |
> > > |-------|---------|------|-------|------|------|
> > > | 3layer-gpt2 | LoGAH | 55.88/68.75 | 0.70 | 47.65 | 52.06 |
> > > | | **TAL** | 62.99/76.70 | 2.89 | 46.21 | 63.21 |
> > > | 6layer-gpt2 | LoGAH | 59.07/72.12 | 1.23 | 46.21 | 53.58 |
> > > | | **TAL** | 61.52/73.70 | 3.32 | 47.29 | 62.76 |
> > > | 9layer-gpt2 | LoGAH | 60.78/74.19 | 1.23 | 48.01 | 57.76 |
> > > | | **TAL** | 58.09/68.97 | 1.75 | 48.78 | 62.09 |
> > > | 12layer-gpt2 | LoGAH | 56.13/68.98 | 0.1 | 48.01 | 58.07 |
> > > | | **TAL** | 52.70/60.53 | 3.05 | 49.82 | 61.49 |
> > > | Avg_Acc | LoGAH | 57.96/**71.01** | 0.82 | 47.47 | 55.37 |
> > > | | **TAL** | **58.83**/69.98 | **2.75** | **48.02** | **62.39** |

---

### Official Review · Reviewer_Tfhf · 2024-11-04

**Soundness:** 3
**Presentation:** 2
**Contribution:** 2
**Rating:** 5
**Confidence:** 3

**Summary:**

Authors introduce Task-Aware Learngene (TAL), a framework for predicting model parameters based on task-specific characteristics and desired model scales. TAL is an incremental improvement on top of previous methods like LoGAH and authors demonstrated better results compared to LoGAH in some cases. The motivation for this is to achieve efficient knowledge transfer and adaptability to descendant models of different scales. The paper highlights the importance of high-quality initialization in improving descendant model performance.


---- Replying here as the review period has ended -- Time: Dec 3, 4:15PM PST

Thanks authors for replying to my questions. I have reviewed the answers and gone through other reviewer comments and discussions. I would like to keep my rating primarily because it's still not clear the solid reasoning/ understanding around how  "Task Aware" idea is helping improve representations. More insight into that will definitely help strengthen the paper. Would recommend following up on that in next iteration. Thank you! Good luck

**Strengths:**

Novelty in terms of proposing task and scale specific initialization: TAL introduces a unique approach for multi-task parameter prediction that adjusts model parameters to specific tasks and scales. The use of task-specific computational graphs in the Learngene module is innovative and contributes a fresh angle to multi-task learning and model initialization.

Broad Applicability/ Significance: TAL’s ability to initialize models for varied tasks and handle models of differing scales suggests it could become a valuable tool in multi-task learning and transfer learning research. It addresses an important problem if solved could lead to massive efficiency wins in efficiently transfering knowledge from foundation models to downstream applications.

Extensive experiments and comparison to LoGAH: Authors have done extensive experiments across multiple tasks and shown improvements compared to LoGAH versions. The low dimensional representation of computation graphs generated by TAL for various tasks are quite interesting.

**Weaknesses:**

Limited contribution: It does look like a minor incremental work over existing hypernetwork methods GHN-2, LoGAH where in the main different is adding task-specific layers/information. It is also not very clear from the paper what is the specific "task-specific" information that is added to the learngene that helps improve the downstream model performance.

Explanation of results/experiments: Table 2 and Table 3 has lot of experiments comparing TAL with LoGAH and showing several cases of improvements. The improvements range from slight to large improvements and in some cases negative improvements. Authors can dig deeper into what's the underlying reasoning behind this improvement/regressions.

Shallow Analysis of Catastrophic Forgetting: The paper mentions TAL addresses catastrophic forgetting, but it lacks specific experiments or metrics to evaluate this claim. There is little evidence showing that TAL explicitly mitigates forgetting across tasks in sequential or continual learning contexts.

**Questions:**

More details around the TSL and Task HyperNet layers. What's the specific input to this? How does this help achieve better representation of the computation graph, etc. In cases where the downstream multi task performance regressed, what might have caused this to happen. Adding discussions around these can further strengthen the paper. Looking forward to hear back from authors around these topics.

---

> ### Author Response · Authors · 2024-11-23
> **Response to reviewer Tfhf**
>
> We thank you for your reviews and address your concerns as follows.
> ### Q 1:
> Limited contribution: It does look like a minor incremental work over existing hypernetwork methods GHN-2, LoGAH where in the main different is adding task-specific layers/information.
> ### A 1:
> Our TAL method is based on the Learngene framework and attempts to leverage large-scale pretrained model's knowledge to create descendant models tailored for downstream tasks, which differs from graph hypernetworks (GHN).  In terms of method, we incorporate modules to process task information, enabling a single TAL model to customize models of varying scales for different tasks.
> ### Q 2:
> It is not very clear from the paper what is the specific "task-specific" information that is added to the learngene that helps improve the downstream model performance.
> ### A 2:
> We explained task information in Section 3.2, which is generated by the ancestry model through the average feature extraction of the task images.
> ### Q 3:
> Explanation of results/experiments: Table 2 and Table 3 has lot of experiments comparing TAL with LoGAH and showing several cases of improvements. The improvements range from slight to large improvements and in some cases negative improvements. Authors can dig deeper into what's the underlying reasoning behind this improvement/regressions.
> ### A 3:
> In our experiments (see Table 2 and Table 3), the TAL method achieves significantly higher average accuracy across multiple tasks compared to the baseline methods. Specifically, with initialized parameters, TAL outperforms the comparison methods on 7 out of 9 tasks and across all tested model sizes. A common challenge in multi-task training is that some tasks struggle to surpass the performance of single-task training[1-3]. Here are some intuitive analyses to this situation with TAL:
>
> a. Reasons for Performance Regressions:
> TAL employs a multi-task training approach that, while enabling knowledge sharing, can sometimes lead to negative transfer between tasks. Specifically, for tasks highly similar to ImageNet in terms of data distribution and features, LoGAH v2 optimizes model parameters more effectively through single-task distillation. This targeted optimization allows LoGAH v2 to perform better on these similar tasks. In contrast, TAL’s multi-task setup does not sufficiently optimize these similar tasks, resulting in inferior performance compared to LoGAH v2 in certain cases.
>
> b. Reasons for Performance Improvement:
> For tasks that show significant performance gains, these tasks differ substantially from ImageNet. Consequently, LoGAH v2’s single-task tuning approach is less effective. TAL leverages multi-task sharing to effectively utilize shared knowledge across tasks and employs task embedding mechanisms to filter and inject task-specific knowledge.
> ### Q 4:
> Shallow Analysis of Catastrophic Forgetting: The paper mentions TAL addresses catastrophic forgetting, but it lacks specific experiments or metrics to evaluate this claim. There is little evidence showing that TAL explicitly mitigates forgetting across tasks in sequential or continual learning contexts.
> ### A 4:
> Our work does not aim to address catastrophic forgetting.
> TAL, is designed to customize models of varying sizes for different tasks. Specifically, the descendant models predicted by TAL are for single tasks. In the TAL model, we added specific modules processes task information, which successfully distinguishes different tasks effectively when handling multiple tasks, and there is no catastrophic forgetting problem.
> ### Q 5:
> More details around the TSL and Task HyperNet layers. What's the specific input to this? How does this help achieve better representation of the computation graph, etc.
> ### A 5:
> We introduces TSL is a simple MLP, and Task HyperNet layers are specifically introduced in Formula 2 of Section 3.2 of the original paper.
> To make it easier to understand their functions, in the revised version we added a new formula in Section 3.2 on page 4. The added content as follows:
>
> The task-specific layers apply these bias parameters to the model's computation graph using the following formula:
> \begin{equation}
>     f^G_{\tau} = \gamma_{\tau} \times f^G + \beta_{\tau}
> \end{equation}
>
> **References**
>
> [1] Naveen Arivazhagan, Ankur Bapna, Orhan Firat, Dmitry Lepikhin, Melvin Johnson, Maxim Krikun, Mia Xu Chen, Yuan Cao, George Foster, Colin Cherry, et al. Massively multilingual neural machine translation in the wild: Findings and challenges. arXiv preprint arXiv: 1907.05019, 2019.
>
> [2] Rabeeh Karimi Mahabadi, Sebastian Ruder, Mostafa Dehghani, and James Henderson. Parameter-efficient multi-task fine-tuning for transformers via shared hypernetworks. arXiv preprint arXiv:2106.04489, 2021.
>
> [3]Vandenhende, Simon, et al. "Multi-task learning for dense prediction tasks: A survey." IEEE transactions on pattern analysis and machine intelligence 44.7 (2021): 3614-3633.

---

### Official Review · Reviewer_kNbL · 2024-11-04

**Soundness:** 2
**Presentation:** 1
**Contribution:** 2
**Rating:** 3
**Confidence:** 4

**Summary:**

Existing learngene methods often rely solely on large model architectures, overlooking the specific features of target tasks. This paper introduces a task-aware framework, TAL, which utilizes a graph hypernetwork to predict model parameters while considering both model scale and task-specific information. TAL functions as an end-to-end framework, learning representations and transferring shared knowledge to enable effective parameter initialization. Empirical results show that TAL effectively predicts parameters, with the predicted parameters providing a stronger initialization than previous methods.

**Strengths:**

**Originality:** The authors propose a novel task-aware parameter prediction framework that effectively integrates task-specific information, enabling the predicted parameters to serve as a more effective initialization point.

**Quality:** The experimental results need a better structure to clearly demonstrate the framework’s effectiveness. The heavy use of abbreviations makes the results challenging to read and confuse readers.

**Clarity:** The overall writing of the paper lacks clarity and hard to read, which makes it difficult to understand the proposed approach and its contributions. See below.

**Weaknesses:**

**Reducing Serving Cost:** The abstract mentions reducing serving costs and latency, but the experiments lack clarity on how the TAL framework actually predicts a smaller parameter set to create a more compact model, which would reduce serving costs and latency.

**Learngene Concept:** The concept of "learngene" remains unclear and appears insufficiently differentiated from the standard pretraining-then-finetuning approach. The description in the introduction does not fully convey the concept, and even with Figure 1, it is challenging to understand what a "learngene" is.

**Preliminaries:** The background on Graph HyperNetworks (GHN) is unclear. Lines 128–132 seem just a general training process. Besides, GHN is trained on M different architectures and N data samples, but the nature of these training data samples is not explained. The preliminaries section could be improved by clarifying the input and output formats needed to train a GHN and standardizing the notation.

**Model Scales:** The paper makes a significant claim about handling different model scales in the introduction, but there is little detail or experimental evidence on this. It remains unclear how TAL predicts or adjusts parameters for models of varying scales.

**Presentation:** The experimental results need a clearer, more structured presentation. For instance, starting with an overview of the experiments would provide context, followed by details on the setup, datasets, and baselines. Most importantly, each experiment should include an explanation of its objectives and expected outcomes, helping to guide readers through the data. This approach would make it easier to interpret the results and understand how the numbers and tables support the authors' claims. Besides, the abbreviation should be used properly.

**Questions:**

**Computation Graphs:** The term "Computation Graphs" is introduced but confusing me. Can the authors provide examples?

**Training TAL on a Large Dataset:** It is unclear what a "large dataset" here means. Does this refer to a pretraining dataset containing diverse domains and concepts, or something else?

---

> ### Author Response · Authors · 2024-11-23
> **Response to reviewer kNbL**
>
> We thank you for your reviews and address your concerns as follows.
> ### Q 1:
> The abstract mentions reducing serving costs and latency, but the experiments lack clarity on how the TAL framework actually predicts a smaller parameter set to create a more compact model.
> ### A 1:
> We provide a detailed description of how TAL predicts model parameters of various scales for different tasks in Section 3 (Methods), so we do not repeat this in Section 4 (Experiments).
> ### Q 2:
> The concept of "learngene" remains unclear and appears insufficiently differentiated from the standard pretraining-then-finetuning approach.
> ### A 2:
> We introduce the Learngene concept comprehensively in Section 1 (Introduction). Instead of fine-tuning the entire pretrained model, Learngene inherits a compact component, known as *learngene*, from the pretrained model to create descendant models of various sizes tailored for different tasks. In Section 3 (Method), we explain that in TAL, *learngene* is the encoder of the TAL model and how TAL predicts model parameters of various scales for different tasks. Additionally, in Section 5 (Related Work), we detail the existing works of Learngene.
> ### Q 3:
> The background on GHN is unclear. Lines 128–132 seem just a general training process. Besides, GHN is trained on M different architectures and N data samples, but the nature of these training data samples is not explained.
> ### A 3:
> We provide a introduction to Graph HyperNetworks (GHN) in Section 1 (Introduction). In section 2 (Preliminaries), we formulate the GHN training process in details. In the datasets subsection of Section 4.1 (Experimental Setup), we describe the model datasets (referring to M different architectures) and image datasets (referring to N data samples) used to train the TAL model in our experiments. Additionally, in Section 5 (Related Work), we detail the existing works of GHN.
> ### Q 4:
> The paper makes a significant claim about handling different model scales in the introduction, but there is little detail or experimental evidence on this. It remains unclear how TAL predicts or adjusts parameters for models of varying scales.
> ### A 4:
> In all our experiments, we initialized models of various sizes using TAL and provided detailed results demonstrating TAL's ability to predict parameters for different scales.
> ### Q 5:
> The experimental results need a clearer, more structured presentation. For instance, starting with an overview of the experiments would provide context, followed by details on the setup, datasets, and baselines. Most importantly, each experiment should include an explanation of its objectives and expected outcomes, helping to guide readers through the data. This approach would make it easier to interpret the results and understand how the numbers and tables support the authors' claims. Besides, the abbreviation should be used properly.
> ### A 5:
> We added an overview of the experiments in the revised version on page 5 and 6.  As for the details on the setup, datasets, baselines, objectives and expected outcomes for each experiment, and the meanings of various abbreviations, we clearly provide them in the section 4 (Experiments) of the original paper.
> ### Q 6:
> The term "Computation Graphs" is introduced but confusing me. Can the authors provide examples?
> ### A 6:
> Our paper utilizes computational graphs based on those used in GHN-3 and LoGAH methods. The computation graphs primarily include node\_feat, node\_info, and edges\_embedding. To clarify, we provide an example using the Vision Transformer (\texttt{ViT}) model to illustrate how computation graphs are constructed.
> ```python
> node_feat:
> tensor([
>     [9],   # 'input'
>     [13],  # 'pos_enc'
>     [5],   # 'msa'
>     [12],  # 'ln'
>     [3],   # 'linear'
>     [7],   # 'sum'
>     [3],   # 'linear'
> ])
> node_info:
> [
>     [[0, 'input', 'input', None, False, False]],
>     [[1, 'pos_enc', 'pos_enc', (1, 768, 14, 14), False, False]],
>     [[2, 'msa', 'msa', (1, 768, 14, 14), False, False]],
>     [[3, 'ln', 'ln', (1, 768), False, False]],
>     [[4, 'linear', 'linear', (768, 3072), False, False]],
>     [[5, 'sum', 'sum', None, False, False]],
>     [[6, 'linear', 'linear', (3072, 768), False, False]],
> ]
> edges embedding:
> tensor([
>     [0, 1, 1],  # input -> pos_enc
>     [1, 2, 1],  # pos_enc -> msa
>     [2, 3, 1],  # msa -> ln
>     [3, 4, 1],  # ln -> linear
>     [4, 5, 1],  # linear -> sum
>     [5, 6, 1],  # sum -> linear
> ])
> ```
> Hopefully, this simple example helps clarify the concept.
> ### Q 7:
> Training TAL on a Large Dataset: It is unclear what a "large dataset" here means. Does this refer to a pretraining dataset containing diverse domains and concepts, or something else?
> ### A 7:
> The "large dataset" mentioned in Section 3 refers to a big dataset in a certain field. The TAL model is trained on this dataset to leverage substantial knowledge from the pretrained model. Specifically, in our experiments, we used ImageNet-1K as the large dataset.

---

> > ### Comment · Reviewer_kNbL · 2024-11-25
> >
> > Dear authors,
> >
> > Thanks for your responses. I think the details you provided are needed in the main paper. Here is my follow-up question.
> > 1. About Q1, I think the missing experimental part is the effectiveness when predicting different scales of the model. Then we can validate how well it can reduce serving costs and latency.
> > 2. I saw your A4, can you point me to your detailed results demonstrating TAL's ability to predict parameters for different scales?
> >
> > Thanks!

---

> > > ### Author Response · Authors · 2024-11-25
> > >
> > > We thanks for your response and address your follow-up question as follows.
> > >
> > > ### Q 1:
> > >  About Q1, I think the missing experimental part is the effectiveness when predicting different scales of the model. Then we can validate how well it can reduce serving costs and latency.
> > > ### A 1:
> > > We provide a well-trained TAL model that can initialize parameters of various sizes for different tasks. The TAL parameter prediction process involves only forwarding the learngene and decoder, averaging 0.48 seconds for the test models in Table 2, which allows the TAL model to predict these parameters in under one second and makes the process nearly instantaneous.
> > > As detailed in Section 4.1, models initialized with our TAL model outperforms other initialization methods in both convergence speed and task accuracy.
> > > ### Q 2:
> > > I saw your A4, can you point me to your detailed results demonstrating TAL's ability to predict parameters for different scales?
> > > ### A 2:
> > > As shown in Table 2, we use vit models of different widths and depths, with parameters ranging from 1.5M to 22M. Specifically, n-Ti in the table refers to n layers vit-Tiny, and n-S refers to n layers vit-Small. In Table 1, Table 3 and Table 4, we also tested models of different sizes. Experiments show that all test models initialized with our method converges faster and has higher accuracy than the comparison methods.

---

### Official Review · Reviewer_kKLx · 2024-11-08

**Soundness:** 2
**Presentation:** 3
**Contribution:** 2
**Rating:** 5
**Confidence:** 4

**Summary:**

This paper proposes a new parameter prediction method based on Graph HyperNetworks (GHNs), called Task-Aware Learngene (TAL). The proposed method aims to address the shortcomings of traditional Learngene methods in adapting to flexible scales and task-specific requirements. By incorporating task-specific information and model scale information, TAL predicts the initial parameters for downstream models, thereby enhancing the efficiency and adaptability of model initialization. Experimental results demonstrate that TAL significantly outperforms existing parameter initialization methods, such as random initialization and LoGAH, across various tasks.

**Strengths:**

1. TAL utilizes task-specific features and flexible model scaling for parameter prediction, thereby achieving efficient initialization for different tasks, which significantly enhances the adaptability and performance of the model across various downstream tasks.
2. By using Graph HyperNetworks to encode structural information, TAL can generate downstream models of different scales based on requirements, supporting model flexibility and task customization, which is particularly effective in resource-constrained environments.
3. The paper conducted extensive experiments on multiple datasets (e.g., ImageNet-1K and Decathlon) to validate the effectiveness of TAL.

**Weaknesses:**

1. While the TAL mechanism improves model flexibility and performance, it also significantly increases the computational and resource costs in multi-task scenarios.

2. The comparative experiments in the paper mainly focus on vision tasks, lacking experiments in other domains. Demonstrating similar performance improvements in other fields would enhance the generalizability and persuasiveness of the method.

3. The paper lacks an in-depth discussion on the setting of sampling weights in multi-task training.

4. The method and experiments are inconsitent in motivation. Specifically, the method aims to enhance the expression of gene data, which are more likely to be sequence and graph structure, while the evaluation focus on images that are not corresponds to structure modeling.

5. The core technique is derived from graph hypernetworks directly. However, it lacks of the novel contribution for customize graph hypernetwork for the main task in this work.

**Questions:**

Q1: Could the authors provide a more detailed analysis of the computational cost of TAL during both the training and inference phases, and compare it to other baseline methods such as LoGAH or Random Initialization? Specifically, how much additional training time and resource consumption does TAL require compared to these methods, and how does this translate into efficiency gains for downstream tasks?

Q2: Can TAL mitigate the inherent conflicts among different tasks?

---

> ### Author Response · Authors · 2024-11-23
> **Response to reviewer kKLx**
>
> We thank you for your reviews and address your concerns as follows.
>
> ### Q 1:
> While the TAL mechanism improves model flexibility and performance, it also significantly increases the computational and resource costs in multi-task scenarios.
>
> ### A 1:
> All experiments are run on an NVIDIA RTX 4090. The table below shows the training time for each method in Section 4.2 of the original paper using a single RTX 4090. The modest increase in TAL's training time is acceptable.
>
> | Method    | ImageNet-1K | Decathlon 9-tasks | Total Time   | ImageNet-1K avg acc | Decathlon 9-tasks avg acc |
> |:---------:|:------------|:----------------:|:------------:|:-------------------:|:------------------------:|
> | LoGAH v1  | -           | 19.10 hours      | 19.10 hours  | -                   | 50.08                    |
> | LoGAH v2  | 27.45 hours | 6.42 hours       | 33.87 hours  | 63.97               | 54.17                    |
> | TAL       | 28.47 hours | 7.72 hours       | 36.19 hours  | 64.82               | 62.01                    |
> ### Q 2:
> The comparative experiments in the paper mainly focus on vision tasks, lacking experiments in other domains. Demonstrating similar performance improvements in other fields would enhance the generalizability and persuasiveness of the method.
>
> ### A 2:
> We train TAL-GPT2 on NLP tasks using the same approach employed for vision tasks, enabling parameter prediction for GPT-2 across various scales tailored to different tasks. Currently, the TAL-GPT2 model is still under training.
>
> ### Q 3:
> The paper lacks an in-depth discussion on the setting of sampling weights in multi-task training.
>
> ### A 3:
> The sampling weight $w_{\tau}\$ in section 3.2 of the original paper is determined by the sampling probability $p_{\tau}\$ described in Section 4.1. Specifically, the sampling weight $w_{\tau} \$ is directly proportional to the task sampling probability $p_{\tau}\$ , which indicates each task's likelihood of being selected during training[1]. This ensures that tasks with higher $p_{\tau}\$ have a greater impact on model updates, allowing the TAL model to effectively learn each task's characteristics.
>
> ### Q 4:
> The method and experiments are inconsitent in motivation. Specifically, the method aims to enhance the expression of gene data, which are more likely to be sequence and graph structure, while the evaluation focus on images that are not corresponds to structure modeling.
>
> ### A 4:
> Our TAL method is based on the Learngene framework which inspired by biological gene inheritance, focuses on extracting compact components, known as *learngene*, from a large-scale pretrained model (referred to as ancestry model) and using them to initialize descendant models. In TAL, *learngene* is the encoder of the TAL model. Our goal is to leverage the knowledge of ancestral models to create descendant models tailored for downstream tasks. As shown in Figure 2 of the original paper, our approach does not involve "gene data". Instead, we use model computation graphs as input to the TAL model and train it on various image datasets. This enables the TAL model to customize models of flexible scales for different tasks.
>
> ### Q 5:
> The core technique is derived from graph hypernetworks directly. However, it lacks of the novel contribution for customize graph hypernetwork for the main task in this work.
>
> ### A 5:
> Our TAL method is based on the Learngene framework and attempts to leverage large-scale pretrained model's knowledge to create descendant models tailored for downstream tasks, which differs from graph hypernetworks (GHN).  In terms of method, we incorporate modules to process task information, enabling a single TAL model to customize models of varying scales for different tasks.
>
> ### Q 6:
> Can TAL mitigate the inherent conflicts among different tasks?
>
> ### A 6:
> Yes, TAL effectively mitigates inherent conflicts among tasks. As detailed in Section 3, we designed task-specific layers to handle each task uniquely. Table 2 and table 3 demonstrate that TAL significantly improves multi-task performance compared to the previous single-task LoGAH training. Additionally, the visualization in Section 4.3 shows that the learngene outputs of TAL models at various scales cluster by task, indicating that TAL successfully reduces conflicts between tasks.
>
> **References**
>
> [1] Colin Raffel, Noam Shazeer, Adam Roberts, Katherine Lee, Sharan Narang, Michael Matena, Yanqi Zhou, Wei Li, and Peter J Liu. Exploring the limits of transfer learning with a unified text-to-text transformer. Journal of machine learning research, 21(140):1–67, 2020.

---

> > ### Comment · Reviewer_kKLx · 2024-11-26
> > **Reply to Authors**
> >
> > Thank you for response. Please improve it in your next submission.

---

> > > ### Author Response · Authors · 2024-12-03
> > >
> > > Thanks for your response.
> > >
> > > We conduct extensive experiments by training TAL-GPT2 on multiple datasets: MRPC, COLA, RTE and IMDB. We compare two parameter prediction approaches - LoGAH and TAL - across different scales of GPT2-small models. The results demonstrated that TAL not only reduced training time by more than 50\% compared to LoGAH but also achieved superior performance across test tasks.
> > >
> > > For performance evaluation, we employ task-specific metrics: both accuracy and F1 recall for MRPC, Matthews correlation coefficient for COLA, and accuracy for RTE and IMDB. The experimental results are detailed as follows.
> > >
> > > | Model | Method | MRPC | COLA | RTE | IMDB |
> > > |-------|---------|------|-------|------|------|
> > > | 3layer-gpt2 | LoGAH | 55.88/68.75 | 0.70 | 47.65 | 52.06 |
> > > | | **TAL** | 62.99/76.70 | 2.89 | 46.21 | 63.21 |
> > > | 6layer-gpt2 | LoGAH | 59.07/72.12 | 1.23 | 46.21 | 53.58 |
> > > | | **TAL** | 61.52/73.70 | 3.32 | 47.29 | 62.76 |
> > > | 9layer-gpt2 | LoGAH | 60.78/74.19 | 1.23 | 48.01 | 57.76 |
> > > | | **TAL** | 58.09/68.97 | 1.75 | 48.78 | 62.09 |
> > > | 12layer-gpt2 | LoGAH | 56.13/68.98 | 0.1 | 48.01 | 58.07 |
> > > | | **TAL** | 52.70/60.53 | 3.05 | 49.82 | 61.49 |
> > > | Avg_Acc | LoGAH | 57.96/**71.01** | 0.82 | 47.47 | 55.37 |
> > > | | **TAL** | **58.83**/69.98 | **2.75** | **48.02** | **62.39** |

---

### Author Response · Authors · 2024-11-23
**General response**

We thank all reviewers for their valuable comments. We have revised the paper, according to their comments. All revised parts are marked as blue.
The details of revisions are referred to the following official comments.

---

### Comment · Area_Chair_Docq · 2024-12-03
**End of reviewer-author discussion phase**

Dear reviewers,

As we near the conclusion of the reviewer-author discussion phase, I wanted to kindly follow up to see if you’ve had a chance to review the author responses on your comments. Could you confirm that you’ve read it and, if needed, update your review and scores accordingly?

Thank you for your time and effort!

Your AC

---

### Note · Authors · 2025-01-23

I have read and agree with the venue's withdrawal policy on behalf of myself and my co-authors.